# Family caregiver experience of caring COVID-19 patients admitted in COVID-19 hospital of a tertiary care hospital in Nepal

Prekshya Thapa[1]*, Sami Lama[1], Gayatri Rai[2], Nidesh Sapkota[3], Nirmala Pradhan[1], Roshni Thapa[4], Pratik Uprety[5], Madhur Basnet[6]☯

1 Department of Psychiatric Nursing, College of Nursing, B.P. Koirala Institute of Health Sciences, Dharan, Nepal, 2 Hospital Matron, Nursing Service Administration, B.P. Koirala Institute of Health Sciences, Dharan, Nepal, 3 Department of Psychiatry, Patan Academy of Health Sciences, Lalitpur, Nepal, 4 Department of Community Health Nursing, College of Nursing, B.P. Koirala Institute of Health Sciences, Dharan, Nepal, 5 Medical Doctor, B.P. Koirala Institute of Health Sciences, Dharan, Nepal, 6 Department of Psychiatry, B.P. Koirala Institute of Health Sciences, Dharan, Nepal

☯ These authors contributed equally to this work.
* prekshya.thapa90@gmail.com

**Data Availability Statement:** All relevant data are within the paper and its Supporting Information files.

## Abstract

### Background

Informal caregivers played a significant role in caring for COVID-19 patients during hospitalization in Nepal. This study aimed to understand the experiences of family caregivers attending to their relatives in a COVID-19 hospital in Nepal.

### Methods

A descriptive phenomenological approach was adopted to understand the caregiver's experience in supporting their relatives admitted to the COVID-19 hospital of the BP Koirala Institute of Health Sciences (BPKIHS). Thirteen caregivers of COVID-19 positive patients were purposively selected from April to June, 2022. Face-to-face interviews were conducted at a caregiver residential facility using the interview schedule developed by the department for the purpose of data collection after obtaining ethical clearance from the Institutional review committee of BPKIHS. Data were audio recorded and manually analyzed.

### Results

Among the 13 caregivers, six were male and seven were female. The findings were categorized into five domains: challenges encountered, changes in physical and mental health, changes in roles and responsibilities, positive experiences, and strategies to ease caregiving tasks. Major challenges included financial burdens, communication problems, stigmatization, Insecurity, substandard accommodation, and visitor restrictions. Caregivers reported negative emotions, unmet physical health needs, and shifts in family and occupational roles. Despite these challenges, caregivers attempted to cope positively and acknowledged the efforts of healthcare personnel and other family members. Suggestions for improving caregiving included providing essential medical and basic services within the

**Funding:** The authors received no specific funding for this work.

**Competing interests:** The authors have declared that no competing interests exist.

hospital, enhancing accommodation facilities, establishing proper communication channels, and allowing visitations.

## Conclusions

Caregivers of COVID-19 patients face significant challenges during the caregiving process. Enhancing hospital services, promoting effective communication, fostering positive attitudes, and ensuring a safe environment can facilitate caregiving tasks.

## Introduction

COVID-19 spread rapidly throughout the world, with dire consequences for low -resource countries like Nepal [1]. Nepal reported its index case on January 23, 2020, and implemented strict measures like nationwide lockdown, social distancing, and travel restrictions after the isolation of the second case on 23rd March [2]. The preparedness, readiness, and response status of any country are crucial in identifying, managing, and preventing health crisis such as the COVID-19 pandemic [3]. However, this was not the case for a low-income country like Nepal [4]. The scarcity of resources such as personal protective equipment, health personnel, infrastructure, and other necessary resources tremendously burdened the existing health care services, even during the last wave of COVID-19 [5, 6]. Despite the multiple challenges and a fragile health care system, Nepal struggled to manage the COVID-19 crisis by increasing free testing at public facilities and offering free in-patient COVID-19 health services at designated COVID-19 hospitals, which was noteworthy. However, the efforts to contain COVID-19 were not adequate [7, 8].

The COVID-19 outbreak raised many public mental health concerns. With unknown dimensions, this disease not only disrupted the lives of COVID-19 patients but also their caregivers [9]. Caregiving involves different types of support, such as emotional, physical, and financial support [10]. Due to the lack of formal caregiving services, family members, also referred to as informal caregivers shoulder the major responsibility of caregiving [11]. Informal caregivers of COVID-19 faced their own exposure risks and increased concerns about self-care and health. When the patient with signs of severe COVID-19 needed hospitalization, caregivers experiences a mix of positive and negative experiences including stigma, change in relationships, fear of contracting the disease, psychological stress, visitation restrictions, economic worries, and grieving, making the caregiving task even more challenging [12–14]. Additionally, several studies indicated that caregivers experienced social isolation, chronic stress, psychological distress, and physical and mental health problems exacerbated by the contagious nature of COVID-19 [15–17].

This study was conducted during the last wave of COVID19 in a tertiary level hospital, B.P. Koirala Institute of Health Sciences, which had a 100 bed COVID-19 hospital and served as the referral centre in Province1. The institute provided a separate caregiver residential facility for the caregivers of COVID-19 patients admitted to the hospital. Despite being the only residential facility in Nepal, this caregiver facility was an open shaded structure with no separate provision for males and females and had the bare minimum of facilities.

Phenomenology is a philosophical method of inquiry that enables researchers to understand the fundamental structures of experiences and is useful to explore previously unknown and overlooked experiences [18]. Since, this was the only caregiver residential facility in Nepal, it was crucial to understand the experiences of caregivers. This understanding will help

identify specific areas of intervention to offer greater support to family caregivers during and beyond the pandemic. Therefore, we used this descriptive phenomenology approach to gain an in-depth understanding of caregiver experiences and the challenges of informal caregiving for COVID-19 patients during the pandemic.

## Materials and methods

### Ethics

Ethical clearance was obtained from the Institutional Review Committee of the B. P. Koirala Institute of Health Sciences, Nepal (IRC no. 2176/021). Informed written consent was obtained from each of the participants before enrolling in the study. We followed the consolidated criteria for reporting qualitative research (COREQ) guidelines in this study [19].

### Research team and reflexivity

The research team comprised multidisciplinary members from the hospital, the Department of Psychiatric Nursing, and the Department of Psychiatry, BPKIHS, as well as the Department of Psychiatry, Patan Academy of Health Sciences, Nepal. The lead author, Prekshya Thapa (PT), is a psychiatric nurse with previous experience conducting qualitative studies; she conducted the data collection. The data collection process was assisted by Roshni Thapa (RT), a community nurse, and Pratik Uprety (PU), a medical doctor, who recorded the interviews. The data collection was supervised by Gayatri Rai (GR), the hospital's matron, and its quality was ensured by the other researchers not affiliated with the hospital, namely Sami Lama and Nirmala Pradhan (SL and NP). Madhur Basnet (MB), a language expert and psychiatrist, was involved in translating the interview guide and overseeing the research design, data analysis, and manuscript preparation. Finally, Nidesh Sapkota (NS), who was external to the team, reviewed the data analysis process. None of the research team members were previously known to the participants.

### Study design

This qualitative study used a descriptive phenomenological approach [20] to understand the family caregivers' experiences, as less is known regarding care experiences related to a newly emerging disease (COVID-19).

### Study setting

The study was conducted at the COVID-19 hospital of the B.P. Koirala Institute of Health Sciences. This hospital has 100 beds and was consistently occupied, especially during the surge of omicron variant of COVID-19. As a tertiary referral centre in Province 1, it caters to a diverse population, leading to a heterogeneous sample in terms of caste, ethnicity, age, socio-economic status, ecological regions and other demographic variables.

### Sample population and sampling technique

Thirteen family caregivers of COVID-19 patients admitted to the ward who assumed primary responsibility and stayed for at least one week at the COVID-19 caregiver residential facility were purposively selected from April 15th to June 30th, 2022. Sample size estimation was determined based on a review of the literatures, which typically recommends using a smaller sample size sufficient to answer the main research questions [21, 22]. Sampling continued until no new information emerged and caregivers provided maximum information, ensuring data saturation on the phenomenon. Maximum variation sampling, considering factors such as age,

gender, educational levels, occupational status, and residential location, was used to obtain comprehensive and diverse data [23].

## Data collection

The caregiver was contacted by PT, who explained the purpose of the study, read out the participant information sheet, and obtained the informed consent of the participant. After taking written consent for participation in the study, he or she was asked about socio-demographic information, and pre-structured open-ended questions were asked using an interview schedule guide in a separate room nearby the COVID-19 hospital, maintaining confidentiality and comfort by the lead investigator herself (PT).

A face-to-face interview was conducted to encourage the participants to discuss their experiences of caring for a patient with COVID-19, using the interview guide developed and validated by the research team for this purpose (Table 1). The interview guide was piloted on two of the caregivers of COVID-19 patients who were not admitted to COVID hospital, and questionnaire modifications were made based on the feedback from pilot testing. Physical distancing and precautionary measures were taken into consideration while conducting the interview. The participants were allowed to express their feelings openly, and when necessary, probing questions were asked to encourage discussions and more clarifications.

Each interview lasted 35 to 45 minutes and was audio-taped by RT and PU and later transcribed. Field notes, including non-verbal communication, were taken by RT and PU. Permission was also obtained to audio record the one-on-one interviews. All interviews were conducted in Nepali, the mother tongue of the participants and the data collector.

## Data analysis

Audio recordings of the interviews were transcribed verbatim in Nepali and later translated into English by the lead researcher, PT. Inductive thematic analysis [24] was conducted by identifying and reporting the themes generated from the data. This process involved familiarizing oneself with the data, coding the information, searching for appropriate themes, reviewing the identified themes and defining and naming the themes. MB and PT coded the data, and samples from the transcripts were cross-checked by SL. No software was used for the data

**Table 1. In-depth interview schedule.**

| Domains of Inquiry | Interview Questions | Probing Questions |
|---|---|---|
| Challenges encountered | Reflecting on your personal experiences, what are the challenges you have faced as caregiver attendee, here in caregiver residential facility? <br> What were the most difficult situations for you? | Could you please explain more about the concern you mentioned? <br> How did you feel about that experience? |
| Changes in physical and mental health | Have you experienced any mental and emotional changes as a caregiver, while attending to your relative and staying in this home? <br> Have you experienced any changes in your physical health? <br> Have you experienced any difficulties in your activities of daily living? | Please explain more about that experience. <br> How did you feel about it? |
| Changes in roles, relationships, and responsibilities | How have your family roles/ responsibilities and/or relationships changed after being caregiver of your COVID-19 relative? <br> Was your occupational or work role hampered? | Please explain more about it. |
| Positive experience | Do you have any positive experiences while caring for and attending to your relative here? | Could you explain bit more about that experience? |
| Making caregiving task easier | Going forward, what kind of changes do you think are needed to make the caregiving task easier? <br> Do you have any specific recommendations for the hospital administration? <br> Any recommendations for the health personnel? <br> How can the existing services be improved? <br> What more facilities will be needed? | Please explain more about it. |

analysis. The analyzed themes and categories were shared with research team members (GS, NP, and NS) for their final review. The final themes were discussed among the investigators and were then confirmed by all team members.

Trustworthiness in the study was maintained according to the criteria proposed by Lincoln and Guba [25]. The co-investigator, RT, contacted the participants by phone and reported the findings to ensure that the extracted codes were consistent with their viewpoints. For dependability, the study results were reviewed by a research expert, NS, who was external to the research team. To ensure conformability, peer checks were done by three co-investigators (MB, SL, and NP) to ensure the accuracy of the descriptions and categories. The transferability of data was achieved by the maximum variation of the samples, different levels of age, education, occupation, and family relationship with the patient.

## Results

### Socio-demographic characteristics of the caregivers

Among the 13 caregivers, the six were male and seven were female. The age of the caregiver ranges from 20 years to 65 years. Most of them (n = 9) were married, and had completed secondary education (n = 4,) and had agriculture as their occupation (n = 5) (Table 2).

### Qualitative result findings

We summarized our findings and categorized the themes under five domains: challenges encountered, changes in physical and mental health needs, changes in role, relationship and responsibilities, positive experience and strategies that could ease caregiving tasks (S1 Fig).

**1. Domain of inquiry: Challenges encountered.** *Theme 1*: *Financial challenges*. Almost all the caregivers reported that increased expenses have created difficulties in managing the situation. One of them said that they were not prepared for the expenses *to be this high* and had to manage them by borrowing money. *"Now the expenses for staying for 7-10days are likely to be around 70–80. We came with a little sum of 10–20 thousand as a loan for this seriously ill patient. We are feeling burdened due to financial difficulties."- IDNO.12.* They also expressed that although the treatment inside the COVID hospital is free of charge, the cost of medicines and other necessary supplies has to be covered by themselves. *"Although they say the treatment is free, we have to pay for the expenses of the medicines ourselves."-IDNO.1.*

**Table 2. Socio-demographic characteristics of respondents.**

| Age | Gender | Occupation | Education status | Home address | Marital Status | Relationship with patient |
|---|---|---|---|---|---|---|
| 35 | Female | Unemployed | Primary level | Tarahara | Married | Daughter |
| 26 | Female | Agriculture | Secondary level | Dhankuta | Married | Niece |
| 28 | Male | Agriculture | Secondary level | Khotang | Never married | Son |
| 21 | Female | Teacher | Bachelor level | Biratnagar | Never married | Daughter |
| 37 | Female | Homemaker | Primary level | Barahshetra | Married | Daughter-in-law |
| 35 | Female | Self employed | Can read and write only | Dhankuta | Married | Daughter-in-law |
| 65 | Male | Agriculture | Can read and write only | Bhojpur | Married | Father-in-law |
| 28 | Male | Agriculture | Secondary level | Jhumka | Never married | Son |
| 50 | Male | Agriculture | Primary level | Dhankuta | Married | Son |
| 20 | Male | Student, Part time teacher | Bachelor level | Jhapa | Never married | Grandson |
| 35 | Male | Self employed | Secondary level | Inaruwa | Married | Son-in-law |
| 42 | Female | Home maker | Illiterate | Dharan | Married | Wife |
| 55 | Female | Home maker | Illiterate | Chatara | Married | Mother |

Another caregiver expressed that some investigations are required to be done outside the COVID hospital, which is very expensive and unaffordable. *"They ask for blood tests at the hospital, but they instruct them to do the test outside. The charges are significantly higher when done outside, and now our money is finished."* -IDNO.2. The increased financial burden had additionally hampered caregivers with limited sources of income who came from remote places."*4–5 lakhs have already been spent, including medicines and travels. You see, we have a limited source of income. In the hills, you cannot easily sell your land; people prefer to leave. We have to take loans; need to take loans for treatment."* -IDNO.7.

*Theme 2*: *Communication challenges.* Another major challenge encountered by most of the caregivers was communication problems such as uncooperative caregivers and unreliable information, as well as improper communication from health personnel. One caregiver reported greater difficulty when the other caregivers seemed indifferent and less cooperative. *"Some friends come, but do not inquire about their whereabouts, do not speak even when called; and do not care about anything. It's difficult when this happens."*–IDNO.2.

Some of the caregivers also expressed the feeling that communication from the health personnel was doubtful and unreliable as they were misinformed about the patient's condition. *"Now, it's like this here- one doctor says one thing and another says other thing. There is one doctor, I do not know his name, who scares us a lot when we ask him about the condition of the patient."*–IDNO 4.

*Theme 3*: *Stigmatization and Isolation*: *The challenging social realities faced by COVID-19 caregivers*. Stigma, discrimination, a neglectful attitude, ignorance, and lack of support were also faced by the COVID caregivers. Caregivers expressed feeling unsupported and ignored. Most others feared even being near these caregivers and avoided communication. *One caregiver reported, "They don't support you once you get Covid. Everybody gets scared and runs away. They don't come even when called." -ID NO.2*: *"Now, this one has got Covid, don't go near him. That's what is being done by ignorant ones."*-IDNO 5.

*Additionally*, caregivers expressed concerns about the final death rituals. They mentioned that if the patient dies in the hospital, nobody will care, and they will not be able to perform the necessary rituals. One caregiver explained, *"If the person dies at home, we could perform our rituals. If they die in the hospital, there's nothing we can do. We face stigma, discrimination, and neglect. People stay away and avoid us. It feels like there's no recognition of the dead person, no care, whether it's the death of a person, or a dog, or a cat." -IDNO.7"*.

*Theme 4*: *Gender disparities and safety concerns.* Female caregivers perceived more stress and vulnerability due to mistrust and fear of unknown male caregivers' intentions, as they had to share the same roof in the open shade of the residential facility. One caregiver expressed, as *"What to say, it's very difficult to stay for a female alone. Not all males consider females to be their sisters. Sometimes I feel terrified. When I'm alone, I have to stay awake the whole night. The day before, I was the only female here and could not sleep the whole night."*-IDNO.2.

In contrast, another female caregiver expressed that she felt more secure due to the presence of male caregivers and pointed out that it might be more challenging to be the only caregiver since the facility is located near a jungle area and is not well facilitated. *"We gather courage to stay because there are other males here. Otherwise, there is no facility, and it's really hard for females to stay. Sometimes, when alone, it feels like a jungle and becomes scary. It feels unsafe. I cannot stay alone: it's very quiet, and there isn't much movements." -IDNO.12.*

Fear of theft, looting, and other criminal acts due to the open space and the remote location were some of the caregivers' serious concerns of the caregivers. *"This place is so open; it feels uncomfortable when I'm alone." -IDNO.8.*

*Theme 5*: *Caregivers' struggles under visitor restriction policies.* Most of the caregiver complained that the visitor restriction policy made them unassured and uncertain about the

patient's condition inside the hospital. This created doubt, mistrust, and confusion among the caregivers. *"I am not able to see how my mother is doing inside. My anxious heart fills with doubt, wondering if she might be dead, especially when I cannot see her. Sometimes I feel like taking her home instead, said IDNO1".*

*Theme 6: Substandard caregiver residential facility and services.* Caregiver reported inadequate basic facilities such as restrooms, lighting, water and sanitation. They expressed, *"There is only a toilet, no bathroom; you have to bathe in the toilet only if you need it." -IDNO 3.* Another caregiver stated, *"The toilet has a lock, but it's broken." -IDNO 6.* IDNO.1 mentioned, *"The toilet and bathroom are very dirty. We could clean, but there are no cleaning supplies."–IDNO 1.*

Improper waste disposal and the presence of mosquitoes at night posed health risk for caregiver. One caregiver reported, *"Wastes are disposed of here; no vehicle collects them. Mosquitoes are everywhere because of the waste."–IDNO 4.*

Additionally, caregivers faced stress due to the canteen, lab services, and drug store being located outside the COVID hospital premises. They complained that *we don't get good food here in Dharmashala. We have to go to the market. The available food is not clean; sometimes the vegetables are dirty."-IDNO13.*

Another caregiver mentioned, *"Medical services are far for everyone. There's also a problem with drinking water here." IDNO 6.* Furthermore, caregiver *faced difficulties accessing emergency labs and other services due to lack of lighting. "*

One caregiver shared, *"There should be lighting, but there's none. Yesterday I had to go to the emergency lab at 12:30am, and it's far. The distance is manageable, but without light, it's scary. The path isn't safe, especially without light."- ID No. 8.*

**2. Domain of inquiry: Changes in physical and mental health.** *Theme 1: Emotional turmoil and mental strain.* Almost every caregiver expressed feeling stressed, burdened, and overstretched as a result of caregiving. Caregivers mentioned being physically exhausted and drained, stating, *"There's a lot of exertion; need to go so far to buy medicines and bring meals for patient; there's a lot of tension. There's so much tension, my god."–IDNO5" "We feel anxious when the patient is sicker."- IDNO 2* They expressed a sense of hopelessness as *"I am hopeless now. I shall bear the treatment as long as my condition and finances allow, and then I will take her home if I am not able to bear the costs. Now also, I am sustaining on donated money."-IDNO 1.*

*"The* fear of contracting the virus and uncertainty regarding the patient's condition made caregivers worry. One of them shared, *"I feel tensed about my mother-in-law's condition. My husband is not here; he's abroad, and he's also worried. After all, she is his mother, who gave him birth. I feel worried if she might not get well. There's also the fear if I might get COVID myself."– IDNO 10". "It's scary; it feels like if something happens to a patient. It's scary either way."- IDNO.2.*

*Theme 2: Physical strain and neglected health: The toll of exhaustive caregiving responsibilities.* Almost every caregiver felt that their basic physical needs, such as food, sleep, and rest, were unmet due to exhaustive responsibilities. They expressed that they had to be alert anytime as they could be called from the hospital at any time, which hampered their sleep. *"Now, it's difficult to sleep; they ask for medicine anytime, irrespective of night or daytime, and cannot sleep just like that." -IDNO 1.*

Headaches, backaches, and fatigue were also reported by caregivers as a result of overexertion and lack of rest. *"I am feeling exhausted with a headache and body ache."–IDNo.11.* Because of inadequate facilities near the COVID hospital, caregivers had to run here and there for necessary supplies and related investigations, making them physically drained and restless. *"It's hustle and bustle for two people; they are calling at night and calling during daytime as well."-ID No. 1. "You have to go here and there now and then."-IDNO 7.*

**3. Domain of inquiry: Changes in roles and relationships.** *Theme 1*: *Burden of altered roles*: *caregivers' struggles with evolving family responsibilities and guilt feelings.* Caregivers have reported that they are unable to fulfil family responsibilities due to their caregiving duties. One caregiver expressed her struggles, saying, *"It's difficult to stay here, leaving my children behind at home. One has college in the morning and another has school in the afternoon, so there's a problem regarding who will prepare the meals. They end up eating the meals prepared in the evening the next morning as well, and that's a source of stress."-IDNO 10.*

Another caregiver shared her feelings of guilt, as she could not properly care for her COVID-positive mother due to the contagious nature of the disease. She explained, **"***Yesterday, I was allowed to meet mom. She went out without a mask to bask in the sun, and I had to move away when she tried to come near me, which made me very uncomfortable. I could not make her hair help her with her dresses while sitting beside her. I felt like she approached me thinking I would, but I had to keep my distance due to the fear of COVID-19, and that made me feel somewhat bad." -IDNO 1".*

*Theme 2*: *The impact of caregiving on employment and livelihoods.* Caregivers also expressed that their caregiving role has hampered their work roles and responsibilities and resulted in financial losses. *One participant shared, "My business is also facing financial losses. There are losses here and there as well. It would be okay, at least if my relative gets well."-IDNO3. My husband was working at the site when suddenly my mother tested COVID positive for COVID-19, so he had to leave his work and come here" -IDNo.1.*

**4. Domain of inquiry: Post-COVID resilience and positive adaptation: Navigating challenges, finding strength, and acknowledging support.** Despite the challenges, COVID caregivers also expressed positive experiences, such as normalizing COVID-19, adapting and positively coping, appreciating the hospital facilities and services, and acknowledging the support from family.

*Theme 1*: *Post Covid concerns and positive adaptation.* Several caregivers voiced post-COVID concerns, particularly regarding the risk of infection after receiving the COVID-19 vaccination. One caregiver shared a personal experience, stating, "Two months after vaccination, I fell ill and remained bedridden, affecting my daily activities" (IDNO11). Similarly, another caregiver expressed apprehension, mentioning, "I heard that vaccination can lead to infection, and my mother indeed contracted the virus after being vaccinated."-IDNO.10.

Furthermore, caregivers acknowledged a decline in the intensity of COVID-19-related stigma in society. One participant (IDNO6) remarked, "*Previously, COVID-19 was highly stigmatized, although it still carries some stigma, it is not as severe as before.*"-IDNO5.

Caregivers have normalized their experience with COVID19, as stated by IDNO6. *"After understanding it better this time, I felt COVID-19 is not as dangerous as we thought. It's not that dreadful a disease, provided we exercise and take care of our diet. It's our own weakness to perceive it that way."-IDNO 6.*

Some of the caregivers also expressed their positivity even in difficult situations, acknowledging that there is no other alternative but to stay positive. *"Whatever it may be, we should handle it in a good way, keep laughing and talking as mentioned by ID NO. 13.* This observation highlights the evolving social perceptions surrounding COVID-19, indicating a shift in public attitudes while underscoring persistent concerns and misconceptions among caregivers.

*Theme 2*: *Appreciation of services and support.* A caregiver expressed satisfaction with hospital services and the care provided by healthcare providers, despite the discomfort of staying in caregiving home. *They stated,—"Although there is difficulty in staying here, the way they are taking care of the patients and treating them is satisfactory; it's good."–IDNO10."* Another caregiver mentioned, *"They have made this facility, and so we need not wait outside, need not sit in*

*the sun. We are staying here like a family, and it's fine. We do get a bit cold, but it's OK. When we have no other option, we need to be satisfied with whatever we have."-IDNO 11".*

*Yet* another caregiver appreciated the opportunity to visit the patient inside the hospital, saying, *"My patient is inside, and they allowed one visitor to go in today. We were concerned about the condition inside, but we saw that our patient was OK. We appreciate this."-IDNO4".* One caregiver acknowledged the support from their family and expressed that *"I do have my family responsibilities, but my husband is helping me in this".–IDNO 2.*

**5. Domain of inquiry: Strategies that could make caregiving task easier.** Nearly every caregiver emphasized the need for improved hospital facilities and services to facilitate caregiving tasks. They highlighted the importance of maintaining hygiene and sanitation in caregiving, which is crucial for managing health and preventing illness. Similarly, caregivers expected proper communication and a positive attitude from healthcare providers to support them in their caregiving roles.

*Theme 1*: *The need for essential services within hospital premises for caregivers.* Many caregivers emphasized the importance of having essential facilities like an emergency lab, medical store, bill counter, and canteen facilities within the hospital premises. One caregiver stated, *"It's quite far if you need to ask for medicines. It should be near as much as possible."-IDNO1 "The issue with getting medicines is that it's far; it would be better if nearer."- ID No2.*

IDNO 5 added, "The drug store should be nearby." IDNO 6 suggested, "It would be easier if you owned a pharmacy! It would be better if we did not have to go far to get medicines; they should be available here." IDNO 7 emphasized the need for a pharmacy within the hospital premises, especially for outsiders, stating, "There should be a pharmacy here so that we don't have to go elsewhere at night. People from outside might not know the place and could end up going in the wrong direction. The facilities should be right here".

Additionally, another caregiver emphasized the importance of having the bill counter and canteen facilities nearby. IDNO 5 said, "The bill counter should be close to the hospital, and the canteen should provide healthy and affordable services. We shouldn't have to go far to pay bills." IDNO 8 echoed this sentiment, stating, "We have to travel quite a distance to get meals; it would be much better if everything we need was available around here".

Additionally, another caregiver emphasized the importance of having the bill counter and canteen facilities nearby. IDNO 5 said, *"The bill counter should be close to the hospital, and the canteen should provide healthy and affordable services. We shouldn't have to go far to pay bills." IDNO 8 echoed this sentiment, stating, "We have to travel quite a distance to get meals; it would be much better if everything we need was available around here".*

*Theme 2*: *Promoting safety and comfort*: *Reconstruction and maintenance of facilities for secure and hygienic caregiving environments.* Caregivers expressed that they faced multiple challenges due to the open caregiver residential facility. The open facility should be made more secure in order to ensure caregivers feel comfortable. Caregiver also mentioned that the existing hospital facilities need maintenance, and that open spaces should be partitioned for the safety and comfort of the caregivers. One caregiver stated, *"It would be better if the sleeping place is well organized. This area is open and needs to be organized. It would be better to have separate rooms with locking doors." -IDNO 5.*

Additionally, concerns were raised about the adequacy of the existing caregiving home, especially during surges in cases. *"This facility is okay for the time being, but if the COVID cases suddenly surge, this space won't be enough. It's better to have one more building. Also, lots of ants come here, and it's better to get it plastered." -IDNO 1*Maintaining sanitation and hygiene was also highlighted by caregivers. They suggested, *"It would be better if the dustbins are emptied daily. Also, separate toilets for ladies and gents would be better." -IDNO 3".*

*Theme 3*: *Effective communication and positive attitude*. Caregivers expect clear communication from health personnel, which would assure them regarding patient condition. One caregiver expressed, *"When we hand over our meals to our patients (visitors usually provide their own meals even though the hospital provides them), we should be clearly informed whether our dear ones have eaten or not. Sometimes they just say that they did not eat, but we should also be told why they did not eat. We cannot have trust if we are not communicated well, as we have not seen inside." -ID NO 4"*.

Caregivers also expect a positive attitude from health personnel, which would make them feel better. They expressed, *"It would be better if we were seen positively when we went there. I hope we are not viewed negatively as caregivers of such sick people. I wish doctors treated our patients without negligence, as we have entrusted our patients to their care, and we would be grateful if they were treated well."-IDNO 6*.

*Theme 4*: *Visitation allowance to provide comfort and support*. Some caregivers have expressed their opinion that they should be compulsorily allowed to visit their patients inside the hospital, which would significantly improve the patient's well-being. *One caregiver stated, "There should be one to two visitors allowed for each patient here. They should be nearby, inside with the patient. It feels meaningless when the patient is inside and we are out here. The patient would feel comforted by seeing her family members and would have high morale. She would also stay there satisfied. We could also ask her what she needs and wants to eat."–IDNO 7*.

Another caregiver emphasized, *"My point is that at least one person should be allowed to stay with the patient." -IDNO5*. Some caregivers also expressed that the presence of a caregiver comforts the patient and improves their overall well-being. One of them stated, *"Whatever the case, we want to go in and see the patient. It's ok, even if allowed once a day. Patients also feel bad if relatives don't visit. They would feel comforted if we visit."-IDNO 6*.

## Discussion

The present study aimed to explore the experiences of caregivers residing in COVID-19 caregiver residential facilities while supporting their hospitalized relatives. The findings were summarized and categorized into five domains of inquiry. Informal caregivers play a vital role in in low- and middle- income countries like Nepal [11]. However, they face numerous challenges in providing care [26].

Our findings revealed various difficulties, including financial challenges, communication issues, stigmatization, insecurity, visitation restrictions, and problems related to substandard caregiver accommodations. Almost every caregiver reported financial burdens consistent with the other s*tudy findings [12, 27] likely due to intermittent lockdowns causing unemployment, especially for daily wage-based labourers and vulnerable populations like disadvantaged and marginalized groups [28, 29]. Additionally, caregiving responsibilities often prevented them from fulfilling work duties, leading to job loss or a lack of payment for daily wage workers.

Nepal's already overstretched healthcare system was ill-equipped to handle the crisis, resulting in increased treatment costs and substantial out-of-pocket expenses [27]. Female caregivers felt unsafe due to shared accommodations, with one caregiver expressing fear as a sole woman among male caregivers.

Concerns about theft and personal safety added to their emotional burden. To address this, we recommend separate male and female rooms with secure locking systems in caregiver facilities for safety and peace of mind.

Our study, consistent with others [30, 31], highlighted caregivers' experiences of stigma, discrimination, neglect, and lack of support from relatives, the community, and health care personnel. Visitation restrictions and communication gaps exacerbated caregivers' uncertainty

about patients' conditions, leading to negative emotions such as doubt and mistrust, consistent with other studies [32]. This emphasized the need for clear communication and information dissemination about hospital policies and patient status [32].

The open and shared caregiver accommodations posed challenges in maintaining basic hygiene, contributing to caregiver distress. Improper waste disposal, inadequate facilities, and distant services added to physical exhaustion and burden.

The study identified changes in caregivers' physical and mental health needs. Constant vigilance due to shared accommodations, fear of contracting the virus, financial burden, unfulfilled roles, stigma, and inefficient management led to negative emotions such as fear, anxiety, helplessness, and burnout, consistent with previous studies [33–35]. Early psychological intervention is crucial to promoting emotional release and mental well-being among caregivers during epidemics.

The study also highlighted the burden of assuming additional responsibilities and adjusting to a new caregiving routine, which is consistent with other studies findings [36, 37]. Female caregivers, particularly in Nepal, faced challenges balancing caregiving with household responsibilities, while male caregivers found their occupational roles affected [38]. Addressing discriminatory gender norms is essential to encouraging equal sharing of caregiving responsibilities [39].

In conclusion, our findings emphasize the need for targeted support, improved communication, and gender-sensitive policies to alleviate the challenges faced by caregivers in similar contexts.

## On the positive note

On a positive note, this study was conducted during the omicron wave of COVID-19 where the cases spiked in a very short duration [40]. However, the public was more aware during this phase of the pandemic and had somewhat adapted to the difficult situation created by the pandemic. Some of the positive experiences were as follows: they had adapted positively to the COVID-19 situation and were appreciative of the governments and health personnel's efforts, as well as the existing facilities of the hospital. Caregivers also acknowledged the support from relatives, the community, and the nation, which can be seen as silver linings and is consistent with other study findings [30, 41].

## Recommendations from the caregivers

This study has unveiled several key factors that can improve the challenges faced by caregivers during the health crisis, based on the experiences of those attending COVID-19 hospitals. Providing amenities such as basic health facilities, medical stores, bill counters, and canteen services within the hospital premises can substantially reduce the need for caregivers to travel, thereby minimizing physical exhaustion and burden.

Moreover, caregivers stressed the urgency of restructuring the existing open accommodation facilities into separate cabins equipped with locking systems to ensure their safety. Caregivers reported feelings of fear, vulnerability, and physical threats due to the open and distant accommodation setup. Hence, they suggested the establishment of a secure environment within the hospital premises, along with maintaining proper sanitation and hygiene in the caregiving area, to promote both physical and emotional well-being among caregivers. Additionally, caregivers emphasized the importance of health personnel displaying a positive attitude and practicing clear communication, echoing findings from previous studies [42, 43]. Another vital suggestion was to permit at least one visitor to see their ill relatives inside the COVID-19 hospital every day. Caregivers highlighted that such visits could enhance patient

comfort, allowing them to express their emotions more freely. Moreover, caregivers themselves would find solace in being able to see their relatives, contributing to their overall satisfaction and well-being.

## Strengths and limitations

This study was conducted in the only caregiver residential home affiliated with a COVID hospital operated by the government in the eastern region of Nepal, providing a distinctive perspective on caregiving experiences. While existing literature has delved into the challenges faced by family caregivers of COVID-19 patients, our research is pioneering in its focus on caregivers residing within a dedicated caregiving facility within a COVID hospital.

However, there are notable limitations to our study. Firstly, despite our best efforts to ensure diverse participant selection, the findings may not encompass the full range of experiences prevalent among the wider population. Future research could explore contextual factors such as income, age, religion, education, social support, and residential status, shedding light on the varied experiences of caregivers of COVID-positive patients.

Secondly, a key limitation is the timing of our research, conducted during the omicron variant of COVID-19. At this point, society had somewhat normalized and adapted to the crisis situation, potentially influencing the caregivers' experiences. It is crucial for subsequent studies to investigate the earlier waves of the pandemic, capturing the unique challenges faced during the initial stages when the situation was more acute.

Furthermore, our study highlighted that caregiver dealing with elderly patients, postpartum mothers, or those facing comorbid illnesses experienced heightened stress and burden. To gain a comprehensive understanding, future research should specifically focus on exploring the additional caregiver burden associated with patients having comorbid conditions alongside COVID-19. Addressing these limitations can contribute significantly to the depth and breadth of knowledge in this critical area of study.

## Implications to policy and practice

Implementing a range of initiatives can significantly enhance the caregiving environment for COVID-19 patients in hospitals. Financial support mechanisms, including insurance coverage and subsidies, can be tailored to ease the economic burdens faced by informal caregivers. Healthcare professionals can undergo comprehensive communication training, ensuring empathetic and clear interactions with caregivers. Public awareness campaigns targeting the reduction of stigmatization against both COVID-19 caregivers and patients can be launched. Hospitals can allocate funds to improve accommodation facilities, focusing on security and hygiene. Visitation policies can be reviewed to strike a balance between patient safety and the emotional well-being of caregivers, possibly incorporating flexible hours or virtual options. Initiatives like formal recognition programmes and awards can acknowledge exemplary caregiving efforts, fostering pride and motivation among caregivers. Research collaboration with institutions can inform policies, ensuring ongoing improvement, while national guidelines for crisis preparedness, including caregiver training, can enhance their confidence and efficiency during emergencies. These concerted efforts can create a compassionate and supportive caregiving environment within COVID-19 hospitals in Nepal.

## Conclusion

This study demonstrates that family caregivers attending to COVID-19 patients in hospitals and residing in caregiver residential facilities face numerous challenges. These caregiving challenges not only affect their physical health needs and activities of daily living but also lead to

emotional and psychological issues such as fear, anxiety, and stress. Stigma and communication problems further exacerbate the caregivers' conditions. However, amid these challenges, caregiver also reported positive experiences of coping and adapting to the difficult situation, as well as appreciating the existing facilities of the hospital.

Our study revealed several key factors that can facilitate care, including improving hospital services and facilities, maintaining a positive attitude and effective communication, providing allowances for visitation to support the patients, and ensuring sanitation in caregiving homes. These facilitators could play a crucial role in managing future health crises.

## Supporting information

**S1 Fig. The concept mapping of themes according to domains of inquiry.** The arrow depicts the way in which each domain is related with other.
(TIF)

**S1 Appendix. Participant informed consent.**
(DOCX)

**S2 Appendix. Consolidated criteria for reporting qualitative studies checklist.**
(DOCX)

## Acknowledgments

We would like to acknowledge the family caregivers who participated in the study. We would also like to acknowledge the support of B.P. Koirala Institute of Health Sciences, College of Nursing, and Department of Psychiatry, for facilitating in smooth carrying out of the research.

## Author Contributions

**Conceptualization:** Prekshya Thapa, Sami Lama, Madhur Basnet.

**Data curation:** Prekshya Thapa, Madhur Basnet.

**Formal analysis:** Prekshya Thapa, Sami Lama, Nidesh Sapkota, Madhur Basnet.

**Investigation:** Prekshya Thapa, Roshni Thapa, Madhur Basnet.

**Methodology:** Prekshya Thapa, Sami Lama, Gayatri Rai, Nidesh Sapkota, Nirmala Pradhan, Roshni Thapa, Pratik Uprety, Madhur Basnet.

**Project administration:** Prekshya Thapa, Roshni Thapa, Pratik Uprety, Madhur Basnet.

**Resources:** Prekshya Thapa, Roshni Thapa, Pratik Uprety, Madhur Basnet.

**Software:** Prekshya Thapa, Madhur Basnet.

**Supervision:** Gayatri Rai, Nidesh Sapkota, Madhur Basnet.

**Validation:** Prekshya Thapa, Sami Lama, Nidesh Sapkota, Nirmala Pradhan, Madhur Basnet.

**Visualization:** Prekshya Thapa, Madhur Basnet.

**Writing – original draft:** Prekshya Thapa, Sami Lama.

**Writing – review & editing:** Prekshya Thapa, Sami Lama, Gayatri Rai, Nidesh Sapkota, Nirmala Pradhan, Roshni Thapa, Pratik Uprety, Madhur Basnet.

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
