## [Decision Letter · Decision Letter 0]

18 Sep 2023

PONE-D-23-21744Family Caregiver experience of caring COVID-19 patients admitted in COVID-19 hospital of a tertiary care hospital in Nepal.PLOS ONE

Dear Dr. Thapa,

Thank you for submitting your manuscript to PLOS ONE. After careful consideration, we feel that it has merit but does not fully meet PLOS ONE’s publication criteria as it currently stands. Therefore, we invite you to submit a revised version of the manuscript that addresses the points raised during the review process.

I would like to suggest authors to review the data and create new themes applicable for post COVID situation and also discuss on limitations and the value added by the study. 

We look forward to receiving your revised manuscript.

Kind regards,

Kshitij Karki, MPH, MA

Academic Editor

PLOS ONE

2. Please include a copy of Tables 1 and 2 which you refer to in your text on pages 8 and 10.

3. We notice that your supplementary figures are uploaded with the file type 'Figure'. Please amend the file type to 'Supporting Information'. Please ensure that each Supporting Information file has a legend listed in the manuscript after the references list.

Additional Editor Comments:

Thank you for conducting this study. Many literature are there in Nepal and internationally regarding this issue. Please go through your data and try to generate the new themes and also consider the post covid situation as the reviewer questioned. I would like to see the limitations as well for the study and what the study will add the value in the evidence / or policy.

Reviewers' comments:

Reviewer's Responses to Questions

**Comments to the Author**

1. Is the manuscript technically sound, and do the data support the conclusions?

Reviewer #1: Yes

Reviewer #2: Yes

2. Has the statistical analysis been performed appropriately and rigorously? 

Reviewer #1: Yes

Reviewer #2: No

3. Have the authors made all data underlying the findings in their manuscript fully available?

Reviewer #1: Yes

Reviewer #2: Yes

4. Is the manuscript presented in an intelligible fashion and written in standard English?

Reviewer #1: Yes

Reviewer #2: Yes

5. Review Comments to the Author

Reviewer #1: Authors has done novel work following guidelines for conducting this qualitative study and writing manuscript.

I have couple of comments:

1. Abbreviations have been used in methodology without mentioning full word such as PT, RT, PU, GR, SL and NP, MB, NS.

2. language editing of text is required

Reviewer #2: Although this study provides useful information about the experiences of informal caregivers of Covid-19 patients, it does not offer unique experiences that add to the qualitative findings of previous studies in this regard. The methodology and findings of this study are similar to the previous studies conducted in this field, and the authors did not mention much innovation in the study. The themes and sub-themes obtained in this study are almost identical to those found in previous studies. Given the time that has passed since the Covid-19 pandemic and the time of conducting this study, it was expected that more different data would be obtained.

6. PLOS authors have the option to publish the peer review history of their article (what does this mean?). If published, this will include your full peer review and any attached files.

Reviewer #1: **Yes: **RANO MAL Piryani

Reviewer #2: No

---

## [Author Response · Author response to Decision Letter 0]

31 Oct 2023

Comment: Please ensure that your manuscript meets PLOS ONE's style requirements, including those for file naming.

Response: Thank you for the feedback. We have made necessary corrections as per the PLOS style requirements. 

Comment: Please include a copy of Tables 1 and 2 which you refer to in your text on pages 8 and 10.

Response: Thank you for the feedback. The copy of Table 1 and 2 has been added to the text(Section: Data Collection Page No.8)

Comment: We notice that your supplementary figures are uploaded with the file type 'Figure'. Please amend the file type to 'Supporting Information'. Please ensure that each Supporting Information file has a legend listed in the manuscript after the references list.

Response: Thank you for the feedback. We have amended the file type figure to supporting information.

Comment: Thank you for conducting this study. Many literature are there in Nepal and internationally regarding this issue. Please go through your data and try to generate the new themes and also consider the post covid situation as the reviewer questioned. I would like to see the limitations as well for the study and what the study will add the value in the evidence / or policy

Response: Thank you for this constructive feedback. We have added a new themes and limitations as well as implication of the study.Section: Strengths and Limitations Page No.27 and 28

Abbreviations have been used in methodology without mentioning full word such as PT, RT, PU, GR, SL and NP, MB, NS. 

Response: The full name has been mentioned at first before using as abbreviated form.Page no.5 

Section Research team and reflexivity

Comment: Language editing of text is required.

Response: The editing of the manuscript has been done as necessary. 

Comment: Although this study provides useful information about the experiences of informal caregivers of Covid-19 patients, it does not offer unique experiences that add to the qualitative findings of previous studies in this regard. The methodology and findings of this study are similar to the previous studies conducted in this field, and the authors did not mention much innovation in the study. The themes and sub-themes obtained in this study are almost identical to those found in previous studies. Given the time that has passed since the COVID-19 pandemic and the time of conducting this study, it was expected that more different data would be obtained.

Response:Thank you so much for the feedback. We have made some changes in themes and sub-themes and had added the unique contribution of the study to policy and practice.(Page no. 28

Section: Implications to Policy and Practice.)

---

## [Decision Letter · Decision Letter 1]

20 Nov 2023

Family Caregiver experience of caring COVID-19 patients admitted in COVID-19 hospital of a tertiary care hospital in Nepal.

PONE-D-23-21744R1

Dear Dr. Prekshya Thapa,

We’re pleased to inform you that your manuscript has been judged scientifically suitable for publication and will be formally accepted for publication once it meets all outstanding technical requirements.

Kind regards,

Kshitij Karki, MPH, MA

Academic Editor

PLOS ONE

Additional Editor Comments (optional):

Reviewers' comments:

Reviewer's Responses to Questions

**Comments to the Author**

1. If the authors have adequately addressed your comments raised in a previous round of review and you feel that this manuscript is now acceptable for publication, you may indicate that here to bypass the “Comments to the Author” section, enter your conflict of interest statement in the “Confidential to Editor” section, and submit your "Accept" recommendation.

Reviewer #1: All comments have been addressed

2. Is the manuscript technically sound, and do the data support the conclusions?

Reviewer #1: Yes

3. Has the statistical analysis been performed appropriately and rigorously? 

Reviewer #1: Yes

4. Have the authors made all data underlying the findings in their manuscript fully available?

Reviewer #1: Yes

5. Is the manuscript presented in an intelligible fashion and written in standard English?

Reviewer #1: Yes

6. Review Comments to the Author

Reviewer #1: Authors have almost addressed all the comments. They may include limitations of the study if any all

7. PLOS authors have the option to publish the peer review history of their article (what does this mean?). If published, this will include your full peer review and any attached files.

Reviewer #1: **Yes: **RANO MAL Piryani

---

## [Editor Report · Acceptance letter]

29 Dec 2023

PONE-D-23-21744R1 

PLOS ONE

Dear Dr. Thapa, 

I'm pleased to inform you that your manuscript has been deemed suitable for publication in PLOS ONE. Congratulations! Your manuscript is now being handed over to our production team.

Kind regards, 

on behalf of

Dr. Kshitij Karki 

Academic Editor

PLOS ONE